# The Seismo-Performer: A Novel Machine Learning Approach for General and Efficient Seismic Phase Recognition from Local Earthquakes in Real Time

**DOI:** 10.3390/s21186290

**Published:** 2021-09-19

**Authors:** Andrey Stepnov, Vladimir Chernykh, Alexey Konovalov

**Affiliations:** 1Far East Geological Institute, Far Eastern Branch, Russian Academy of Sciences, 690022 Vladivostok, Russia; director@fegi.ru; 2Khabarovsk Federal Research Center, Far Eastern Branch, Russian Academy of Sciences, 680000 Khabarovsk, Russia; admvc@ccfebras.ru

**Keywords:** seismogram, spectrogram, transformer, attention, CNN, deep learning, seismic phase, real-time automation, classification, computational efficiency, local seismic network

## Abstract

When recording seismic ground motion in multiple sites using independent recording stations one needs to recognize the presence of the same parts of seismic waves arriving at these stations. This problem is known in seismology as seismic phase picking. It is challenging to automate the accurate picking of seismic phases to the level of human capabilities. By solving this problem, it would be possible to automate routine processing in real time on any local network. A new machine learning approach was developed to classify seismic phases from local earthquakes. The resulting model is based on spectrograms and utilizes the transformer architecture with a self-attention mechanism and without any convolution blocks. The model is general for various local networks and has only 57 k learning parameters. To assess the generalization property, two new datasets were developed, containing local earthquake data collected from two different regions using a wide variety of seismic instruments. The data were not involved in the training process for any model to estimate the generalization property. The new model exhibits the best classification and computation performance results on its pre-trained weights compared with baseline models from related work. The model code is available online and is ready for day-to-day real-time processing on conventional seismic equipment without graphics processing units.

## 1. Introduction

Phase picking is a routine task in the processing of local seismological monitoring data. The complete automation of this task has become increasingly important, especially in connection with the growth of seismic networks with inexpensive instruments and the increase in the number of Internet of Things (IoT) devices [1,2].

Phase picking automation is a challenging problem. The number of lower magnitude earthquakes has grown exponentially [3]; however, the amplitudes of many earthquake signals are weakened to the level of seismic noise or less with decreasing earthquake magnitudes. Improving the completeness of the earthquake magnitude catalog is a central goal of local seismological monitoring since a comprehensive catalog provides more information about the seismic regime.

Another issue is the configuration of the seismic network. A wide variety of sensor types, site soil conditions, and levels of seismic noise can exist inside a single network. Of course, this can differ from one network to the next. Consequently, the registered shape of a seismic signal can vary significantly, and a general algorithm is needed to address the phase picking task in a manner meeting or exceeding human effort for data coming from any local seismic network.

Machine learning techniques allow us to fit the parameters of an arbitrary function utilizing existing labeled data to make accurate predictions on new data coming from the same complex distribution. Usually, phase picking automation is treated as a classification task. One approach is to use split wave data on small windows (4–6 s) that contain only one centered pick or noise, e.g., in [4,5]. Each window or sample is considered to belong to one of three classes. The processing of small windows of waveform data is more suitable for real-time seismic systems because ground motion data are received continuously in small chunks. Thus, each successive chunk can be added to the previous contiguous data to form the length of the window and be fed directly into the fitted model for class detection. Since the window length is small, detecting the arrival of P/S times from earthquakes occurs within a few seconds after the arrival of the data with the registered phase. However, the exact detection time can slightly vary depending on the implementation of a sliding window algorithm. For this approach, in most machine learning architectures, convolutional neural networks (CNNs) have the lowest error rates in terms of signal-to-noise discrimination [5]. Moreover, CNN-based models such as the generalized seismic phase detection model (GPD) demonstrate efficiency for classifying local seismic phases at the human performance level [4].

Another approach is to process earthquake length windows (30–60 s) with more complex triggers, e.g., [6,7]. The output of this type of model is a probability distribution over the window length. For example, PhaseNet [7] outputs P/S phases and noise distributions, while the earthquake transformer model (EQT) [6] provides P/S phases and earthquake detection triggers. These models have good accuracy when scanning archives, but this approach is not suitable for real-time processing due to the required long input window length. However, this approach leads to complex models that are difficult to optimize in terms of computational efficiency.

However, the generalization properties of machine learning models have not been well tested or assessed [8]. Intuitively, the shape of seismic signals from local earthquakes must be preserved from one network to another, even if they are affected by noise and other factors. In this work, we aimed to create a model for accurately picking P and S waves from local earthquakes in real time. To test the generalization property, we introduced two datasets that contain P and S waves recorded from local earthquakes in different regions of the earth. That data were not involved in the training process for any model to estimate the generalization property. The models presented in the work have the best indicators of the accuracy of the TOP-1 classification of seismic phases from local earthquakes for test datasets.

Typically, seismic data servers are not equipped with graphics cards or any special hardware. We developed the most efficient model, suitable for use on the central processing unit (CPU) during inference. On GitHub, we provided the model implementation, code examples, helpful tools for scanning seismic archives, and pretrained most suitable weights. In this way, it is possible to implement a phase recognition unit in real-time seismic data processing systems, such as earthquake early warning systems, and in many seismological centers that process data, to quickly inform about recent earthquakes and their impacts.

## 2. Materials and Methods

Recent advances in solving natural language processing problems applied to image classification tasks have shown that transformer architecture with an attention mechanism [9] can outperform state-of-the-art CNN-based models [10]. Moreover, the transformer architecture lacks some of the inductive bias inherent in CNNs [10]; therefore, the transformer architecture is more generalized. To test the performance of the transformer architecture, we developed a new model for phase classification of local earthquakes.

An overview of the model design is shown in Figure 1. We followed the original image recognition model based on transformer architecture [10], with modifications specific to the seismic data. The input sample is 4 s 3-channel 100 Hz waveform data recorded by a local velocity seismograph. A raw signal from a digitizer was normalized by the absolute maximum amplitude observed on any of the three components. All the data were detrended and high-pass filtered above 2 Hz. We removed the original date and time stamps, so each entry started at 0 and had 400 samples in total.

First, the spectrogram was computed as a 2D representation of the raw signal for each channel. To this end, we successively applied the short-term Fourier transform (STFT), calculated the magnitude to obtain the floating-point tensor after a complex STFT operation, and finally, converted the magnitudes to decibels. Then, we normalized the data using the maximum absolute scaler, as this leads to more stable training results. During the STFT operation, seismology-specific parameters were applied (Table 1) to obtain an adequate spectrogram that represents the seismic phases more clearly than the raw signal.

Next, we handled the spectrogram as an image similar to the vision transformer approach [10]. The two-dimensional (2D) spectrogram x=ℝH×W×C was reshaped into a sequence of flattened 2D patches xp=ℝN×(P1·P2·C), where H and W indicate the resolution of the original spectrogram, C is the number of channels, P1 and P2 are the patch sizes along the H and W dimensions, and N=(H·W)/(P1·P2) is the resulting number of patches. We applied rectangular patches to the spectrogram, which differs from the squared regions in the original vision transformation approach, as they lead to a more accurate classification in our specific model.

Subsequently, we mapped the patches to D dimensions with a trainable linear projection to obtain embedded data with a fixed latent vector size as follows:(1)xp:ℝN×(P1·P2·C)·ℝN×D

Next, we added a learnable classification token and position information to the embedded patches as follows:(2)z0=[xclass0,xp1,xp2,…,xpN ]+Epos, Epos∈ℝ(N+1)×D
where xclass0 is the classification token, Epos is the positional information, and z0 is the resulting sequence that contains embedded patches with positional encoding.

The core block of the proposed model is the transformer encoder [9], which consists of L layers following one another in a sequential manner,
(3)$$z_{l}^{\prime }=\mathrm{MSE}\left(\mathrm{LN}\left(z_{l-1}\right)+z_{l-1}\right),\;l=1,\ldots ,L$$
(4)$$z_{l}=\mathrm{MLP}\left(\mathrm{LN}\left(z_{l}^{\prime }\right)+z_{l}^{\prime }\right),\;l=1,\ldots ,L$$
where LN is the layer normalization function, MSE is the multiple head attention, and MLP is the simple feed-forward neural network with GELU [11] nonlinearity. To improve computational efficiency, we replaced the original multiple head attention operation (MSA) with the FAVOR+ mechanism (MSE) [12], which has linear space and time complexity to form performer layers at the end.

Finally, we used the representation state of the classification token that passed throughout the performer encoder to perform the classification step as follows:(5)ℙ(P,S,N)=SoftMax(MLP–head(zL0))
where zL0 is the representation state of the classification token given from the performer encoder output, and MLP–head is the feed-forward neural network with one hidden layer, which has the same configuration to MLP inside performer layers. To treat the outputs of the model as class probabilities, we applied the SoftMax activation function.

To configure the resulting model, we selected a rectangular patch size of P1×P2=22×3, with projection dimensions of D=48, and L=2 consecutive layers inside the performer encoder. The MSE and MLP functions have their own parameters, such as the number of attention heads and dense units. We set 2 heads for MSE. For the MLP and MLP–head hidden layers, we picked an equal density of 96 units. The final Seismo-Performer configuration has 57, 123 trainable parameters (Table 1). To prevent overfitting, we added dropout layers with a constant rate (0.1) between the dense layers. In addition, a relatively small projection dimension (D) is also an overfitting prevention mechanism.

The implementation of the model and its configuration, the pretrained best-fitted weights, the code examples, and documentation are available online (see Data Availability Statement).

To train and validate the model, we used 4.5 million 3-component 4 s seismograms recorded in Southern California [13], as this is the largest available dataset of local earthquakes. To additionally test the model, we used local earthquake data collected from two different regions using a wide variety of seismic instruments. These data were not involved in the training process so as to estimate the generalization property as accurately as possible.

The first region was Sakhalin Island (Figure 2A). This region’s local seismic network extends along Sakhalin and is notably sparse. On average, earthquakes here are localized at only 3 stations. Seismic stations are installed near settlements due to the need for a central power supply and the availability of mobile network coverage. In general, these are noisy stations. This seismic network was originally established to monitor seismic induction in connection with oil and gas production in the northern Sakhalin region. The seismic network is mainly composed of short-period sensors, such as Lennartz LE3D-Lite and OSOP Raspberry Shake. We selected earthquakes that occurred between September 2006 and March 2021 and filtered out weak events below M1.0 or source depths ≥ 50 km. In total, the dataset for Sakhalin after filtration contained 4702 local earthquakes (Figure 3A).

The second region was located in the northern Caucasus in the center of Dagestan (Figure 2B). This seismic network was created to monitor the seismic regime in the vicinity of hydroelectric power plants. Each seismic station is equipped with a Guralp CMG-6T broadband sensor. On average, earthquakes are localized by 6 or more stations. We used data from January 2018 to June 2019 and the same filters used for Sakhalin to obtain 1750 events in total (Figure 3B).

To construct both datasets, we selected only those manually picked phases that were recorded by a station with an epicentral distance of less than 300 km to track only local events. For impulsive seismic noise sampling, a short-term average/long-term average (STA/LTA) filter was applied to continuous data with a trigger factor equal to 3.5. We also verified that the noise was not actually a seismic event. There was insufficient impulse noise for the Dagestan dataset. To prevent class imbalance, we collected more noise samples by picking wave windows starting 5 s before randomly choosing the P-wave pick. The summary information about datasets is shown in Table 2. The Sakhalin and Dagestan datasets presented in this work are available online in HD5 format (see Data Availability Statement).

## 3. Results

To report the results, we trained the Seismo-Performer model several times. First, we randomly split the seismograms from the California dataset (Table 2) into training (80%) and validation (20%) sets. Then, the model was compiled with a cross-entropy loss function and the ADAM optimization algorithm [14]. The model was trained using mini-batches of 480 records and a learning rate of 0.001 on the NVIDIA Tesla P100-SXM2 graphical processing unit for approximately 3.8 h. The training process was terminated after 48 epochs (full iterations through the training dataset) on average since there were no performance increases in the validation data for the last five epochs. After completing the training, we evaluated the fitted model on the Sakhalin and Dagestan test datasets and recorded the results. Then, we randomly partitioned the California dataset again with a different random state, compiled, fit the model, and reported the results using the same procedure as in the first step. The training process was completely stopped after 10 split/train iterations due to the absence of significant variability in the indicators during the validation and testing of the data.

To further evaluate the generalization of the model, we performed the above training procedure on a specific Sakhalin dataset and tested on all the data from California and Dagestan. The training, testing, and validation code is available online as a Google Colab notebook (see Data Availability Statement). We fixed random seeds to provide nearly the same reproducible results, however, they may differ slightly due to the stochastic nature of machine learning algorithms.

To compare the accuracy metrics of the Seismo-Performer model, we applied three suitable models using the same input sample. First, we replaced the Performer block with CNN after the spectrogram to build the Spec-CNN model. The accuracy metrics of the spectrogram-based models were compared with the original CNN-based GPD model, which is the state-of-the-art model for short window length raw seismic signal processing. Since the code of the GPD model has not been updated for 3 years [15], we redeployed this architecture in the latest version of the machine learning framework to speed up the learning and inference times. We designated the reconstructed GPD model as a fixed GPD. All these models were fitted using the same procedure as for the Seismo-Performer model.

To assess the accuracy of the models, the TOP-1 score was used. That measure checks if the top class with the highest SoftMax value matches the true label. The TOP-1 validation and test data accuracy for all the tested models are shown in Table 3. It can clearly be seen that the spectrogram-based models outperformed the GPDs across all validation and testing datasets. The difference in the accuracy of the validation data is less noticeable because it comes from the same dataset as the training data for all models. The distinction of the TOP-1 metric is clearer for the test datasets, which may indicate better generalizability of the Seismo-Performer and Spec-CNN models. This is notable, although when training on the Sakhalin dataset of only 7.84 k samples, the difference in accuracy rates for the test data was much higher (Table 3, Panel B). Spectrogram models better learn the shape of waveforms of local earthquakes even with small amounts of data and, therefore, tend to be more general. To examine the differences in accuracy, we plotted the precision/recall curves of the test datasets for all the models with various prediction thresholds (Figure 4). Metrics are defined as follows:(6)$$Precision_{P}=\frac{TPP_{P}}{TPP_{P}+FPP_{S}+FPP_{N}},$$
where $TPP_{P}$ is a true positive P-phase that is marked correct by the model and matches a human label. $FPP_{S}$ is a false-positive P phase, classified as P, but in reality it is an S phase. Additionally, $FPP_{N}$ is a false-positive P phase classified as P, but it is actually noise class.
(7)$$Recall_{P}=\frac{TPP_{P}}{TPP_{P}+FNP_{S}+FNP_{N}},$$
where $FNP_{S}$ is a false-negative P phase, which is designated by the model as S, but it is P. Additionally, $FNP_{N}$ is a false-negative P phase, which marked a true P as some noise.

The precision/recall metrics for S phase were defined in the same fashion.

We can conclude that for all cases, the Seismo-Performer and Spec-CNN had significantly better recall rates with about the same precision, compared with the GPD. This is a significant finding that is detailed in Section 4.

To assess computational efficiency, we developed a miniSEED [16] archive scanner. This tool processes 24 h, single-channel archive files, similar to those generated by the BUD schema [17], which is a common way of storing continuous data from local instruments. The scanner itself has many configuration options, explained in detail in the GitHub documentation (see Data Availability Statement). The most critical option is offsetting (shift) the 4 s sliding window. We chose 40 ms as the default and we explain our reasoning in Section 4. An example of archive scanning using different models is shown in Figure 5. With this tool, we measured the models’ inference times on a 24 h 3 ch archive with a sliding window shift of 40 ms, using only the central processor unit (Table 3). The Seismo-Performer model was more than twice as fast as the original GPD, and nearly 40% as fast as the fixed GPD. The Seismo-Performer model is the fastest model with almost the same accuracy compared with the Spec-CNN and, therefore, is the most suitable for day-to-day processing.

## 4. Discussion

For the short window length approach, each 4 s sample is considered to belong to exactly one class. The main objective of this study was to accurately pick the arrival times of seismic phases to utilize this information for further routine processing. To this end, we implemented a sliding window method, and the 4 s window was shifted to smaller times, in milliseconds, to form a series of overlapping 4 s samples. The sample with the maximum SoftMax probability for a particular class, exceeding the configured threshold parameter, is selected by the sliding window algorithm as the sample containing the seismic phase (P or S). The final arrival time of a seismic phase is picked by a central point of the selected sample since all picks were centered in the training set. All the models considered in this study accurately pick an arrival time. The accuracy of the picking time is affected only by the window shift parameter. For example, if we set a 10 ms shift, we can center pick more precisely and, hence, can obtain more accurate pick times considerably close to those of human ability. On the other hand, a very small shift results in more counts; for a value of 10 ms, we obtain 858,349 4 s windows in a 24 h seismogram case to feed to a model, which affects the inference time. However, there is some inaccuracy in the predictions of a fitted model, uncertainty in human labeling, time synchronization errors (especially in the case of the Network Time Protocol), etc. Finally, we chose 40 ms as the default optimal shift value (which led to a 4-fold decrease in the number of samples), which does not significantly affect the errors in the localization of an earthquake source.

False positives are problems that occur when testing models with continuous data, especially in noisy stations. This is not a special case for the models proposed in this study and affects all existing models, including PhaseNet and EQT. This issue can be corrected by using the prediction threshold parameter. However, false positives should be distinguishable in terms of SoftMax probabilities and have lower values, compared with true seismic phases. In this case, we can set the prediction threshold high enough to filter out false positives. This is the reason the better recall rate (Figure 4) is so important. After extensive testing of our models on continuous data, we chose 0.9997 and 0.9995 as the default prediction thresholds for the P and S phases, respectively.

In regard to EQT, another state-of-the-art model using the long window approach, since the framework propagates different methods of prediction, we resliced the Sakhalin and Dagestan samples as follows: The records of the earthquakes were sliced to 1 min lengths, totaling 2576 and 6898 samples from the Sakhalin and Dagestan continuous data. To evaluate false positives more precisely, we added the same proportion of samples with impulse noise picks and normal seismic noise from each source of the data. We flagged true positives if the model predicted a phase within ±2 s of the true label and false positives otherwise. The model accuracy was only evaluated on the P and S labels without noise class. We used pretrained weights and the default settings of the EQT framework. Eventually, we achieved 67.97% and 46.02% accuracy for the Sakhalin and Dagestan test sets, respectively. These results are less general (Table 3, Panel A). However, EQT is a very powerful tool, especially it has very low false-positive rates on seismic noise records (less than 0.01%). Better results can be achieved by training this model on local earthquake data only, such as Southern California data [13], and optimizing the prediction hyperparameters (normalization, filtering, thresholds, etc.)

## 5. Summary and Conclusions

The new machine learning models were developed for accurately picking seismic phases from local earthquakes on the level of human capabilities. The models process the signal based on a spectrogram rather than raw waveforms. This method made it possible to reduce the number of trained parameters and build more general models for recognizing seismic arrivals. Using two new test datasets, the best P/S phase prediction accuracy for the new technique is fully demonstrated. The Seismo-Performer is one of the proposed spectrogram-based models that demonstrates the best computational efficiency through the use of the attention mechanism.

Since the models process 4 s input samples, they are suitable for real-time continuous seismic data processing. The issue of false positives that occur when processing continuous data can be completely solved by tuning the hyperparameters. With the supplied models code, pretrained weights, and supporting tools, machine learning processing can be quickly deployed to existing seismic data centers. In the case of Seismo-Performer, only the CPU can be used without any expensive acceleration hardware.

## Figures and Tables

**Figure 1 sensors-21-06290-f001:**
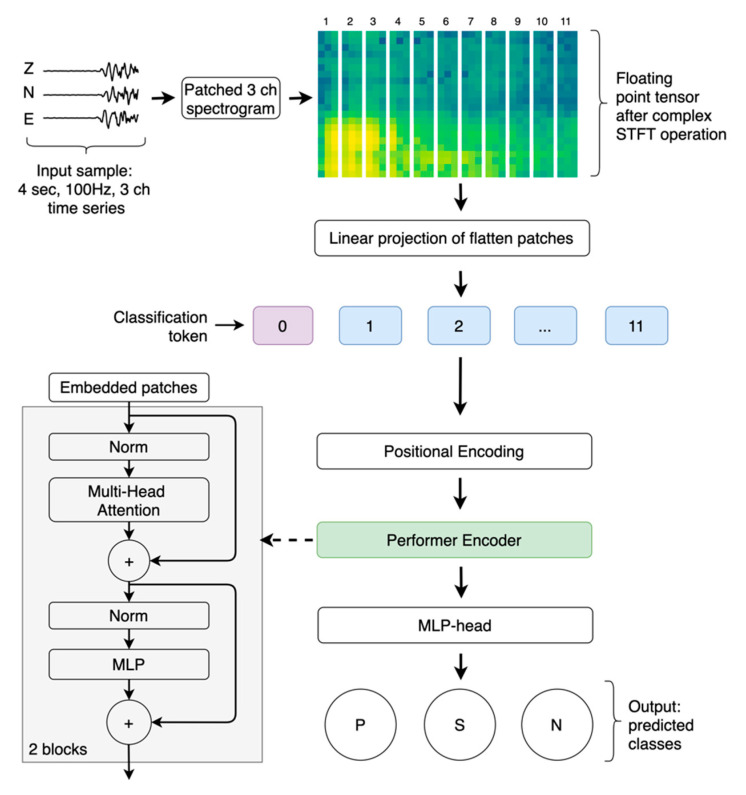
Seismo-performer model overview.

**Figure 2 sensors-21-06290-f002:**
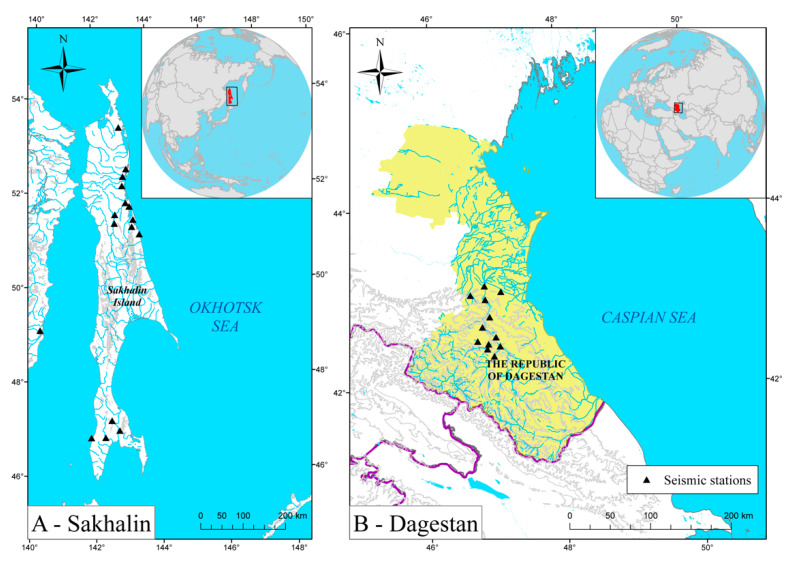
Location of local seismic networks and the geometry of the stations for **Sakhalin (A)** and **Dagestan (B)** regions that were used for the data acquisition.

**Figure 3 sensors-21-06290-f003:**
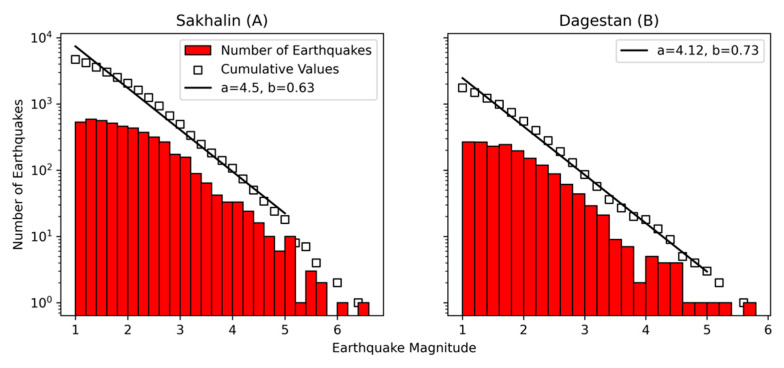
The cumulative Gutenberg–Richter frequency-magnitude distribution of seismic events across **Sakhalin (A)** and **Dagestan (B)** datasets, lines are the best fitting values.

**Figure 4 sensors-21-06290-f004:**
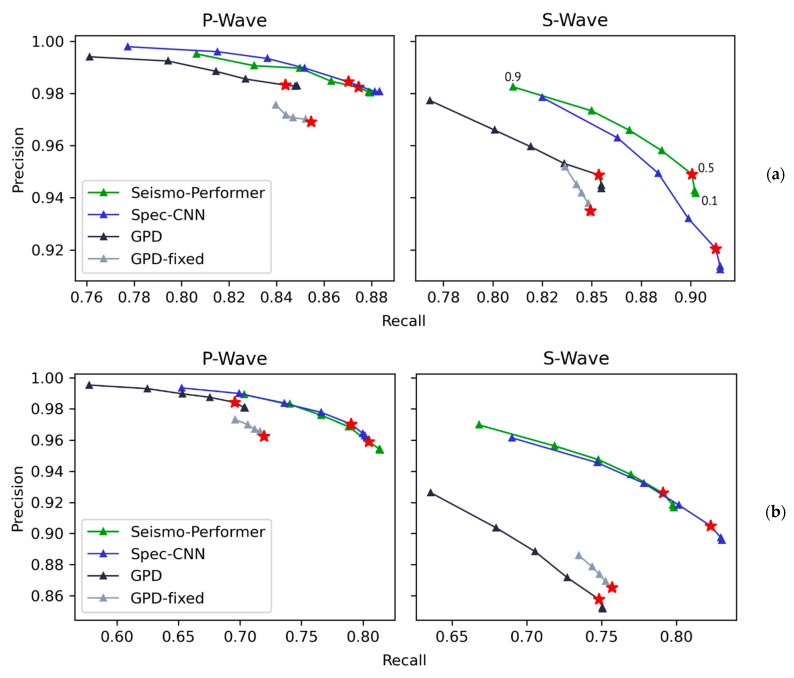
Precision/recall curves for the Seismo-Performer, Spec-CNN, and GPD classifiers on the Sakhalin (**a**) and Dagestan (**b**) datasets. The models were trained on the Southern California dataset only. Precision and recall were computed for different probability thresholds from 0.1 and 0.9 in increments of 0.1, resulting in nine precision and recall values for each classifier/dataset. Red stars show values for a probability threshold of 0.5.

**Figure 5 sensors-21-06290-f005:**
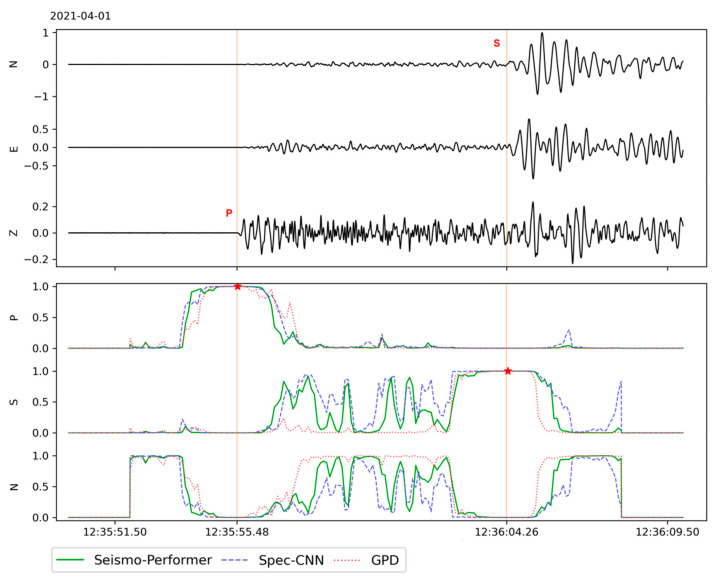
An example of the model’s prediction during the processing of a 1 min seismogram. This is the NYSH seismic station with a 3-channel component “ENZ” (East, North, Vertical) located 97 km from the epicenter of an M3.1 local earthquake on northern Sakhalin [18]. “P”, “S” and “N” indicate the probability (SoftMax values) of predicting P/S phases and noise, respectively. The red vertical lines are the manually picked P and S arrival times. The pentagrams show the maximum SoftMax values for predicting the seismic phase class, which are nearly the same for each model.

**Table 1 sensors-21-06290-t001:** The Seismo-Performer model configuration parameters.

Parameter	Value
Number of FFTs	64
Hop length	16
Spectrogram dims (H, W, C)	22 × 33 × 3
Patch size 1 (P1)	22
Patch size 2 (P2)	3
Number of patches (N)	11
Projection dim (D)	48
Performer layers (L)	2
MLP hidden layer size	96
MLP output layer size	48
Number of attention heads	2
Dropout rate	0.1
Total trainable parameters	57, 123

**Table 2 sensors-21-06290-t002:** Summary of local earthquake datasets. Each sample is 4 s long at 100 Hz (400 points in total). The data were high-pass filtered above 2 Hz, and a trend was removed.

Dataset	Samples Per Class	Total Samples	Data Time Range	Reference
Sakhalin	3.3 k	9.8 k	2006–2021	This work
Dagestan	9.4 k	28.1 k	2018–2019	This work
California	1.5 m	4.5 m	2000–2017	[13]

**Table 3 sensors-21-06290-t003:** Comparison of the TOP-1 accuracy in percent (±std.dev.) and inference times on CPU^1^ in seconds on test datasets. All models were fitted 10 times on the California (Panel A) and Sakhalin (Panel B) train sets with different random seeds. We reserved 20% of the training dataset for validation at each iteration with different seeds.

Train Data (Frac.)	Test Data (Frac.)	Seismo-Performer	Spec-CNN	GPD-Fixed	GPD [4]
Panel A California (0.8)	Sakhalin (1.0)	90.69 ± 0.2	91.43 ± 0.2	89.21 ± 0.5	89.08 ± 0.9
	Dagestan (1.0)	84.57 ± 0.7	85.32 ± 0.4	81.94 ± 0.9	81.11 ± 1.2
	California (0.2) ^2^	98.71 ± 0.01	98.80 ± 0.03	98.63 ± 0.02	98.64 ± 0.06
Panel B Sakhalin (0.8)	Sakhalin (0.2) ^2^	93.71 ± 0.4	95.18 ± 0.6	92.40 ± 1.1	93.37 ± 0.4
	Dagestan (1.0)	83.63 ± 0.7	86.43 ± 0.5	73.03 ± 0.6	73.65 ± 0.5
	California (1.0)	85.59 ± 2.1	85.05 ± 0.8	81.09 ± 1.0	82.31 ± 0.9
Parameters		57 k	176 k	1742 k	1742 k
CPU ^1^ inference time (s) during 24 h 3 ch archive scan with 40 ms shift		56.2 ± 0.3	82.8 ± 0.3	89.7 ± 0.5	123.6 ± 0.4

^1^ Intel(R) Xeon(R) CPU E3-1270 under Linux Kernel-based Virtual Machine (KVM) with 2 virtual cores. 2 20% of the dataset was used for validation during training.

## Data Availability

The implementation of the model and its configuration, the pretrained best-fitted weights, the code examples, and useful tools for continuous archive scanning and documentation are available online at GitHub: https://github.com/jamm1985/seismo-performer (accessed on 18 September 2021). Datasets for Sakhalin and Dagestan are available online in HD5 format at Google Drive (the links are also present on the GitHub page): https://drive.google.com/file/d/1dH2JF9TQmyB6GpIB_dY1jiWAI5uqp6ED (accessed on 18 September 2021), https://drive.google.com/file/d/156w3I9QVnhkCo0u7wjh-c6xekE9f6B3G (accessed on 18 September 2021).

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
