# Peer review of "The Seismo-Performer: A Novel Machine Learning Approach for General and Efficient Seismic Phase Recognition from Local Earthquakes in Real Time"

_sensors, 2021, doi:10.3390/s21186290_

Round 1

Reviewer 1 Report

Since there is a vast seismological literature on the Neural Network supported seismic signal phase recognition - the title should better communicate this novelty of this particular submission. E.g. "Seismo-performer: A novel machine learning approach for ..." etc. The same problem is with the last two paragraphs of Introduction:
Quote: 
(...) "In this work, we aimed to create a model for accurately picking P and S waves from local earthquakes in real time. However, the generalization properties of machine learning models have not been well tested or assessed [8]. Intuitively, the shape of seismic signals from local earthquakes must be preserved from one network to another, even if they are affected by noise and other factors. (...)
Typically, seismic data servers are not equipped with graphics cards or any special hardware. We have developed (...)"
Clearly, the above second sentence lacks logic as it looks as if it criticises the actual, proposed novel method, while it probably deals with ref. [8]. Please rearrange these two paragraphs so that the novelty of the current submission is clearly communicated compare to previous literature, in particular [8] but not limited only to critique of this one position.
This submission lacks "Conclusions". Taking into account a long discussion of the results presented one may expect some conclusions highlighting novel results of this paper. Please add chapter 5 titled Conclusions or "Summary and conclusions" or in a similar way.

Detailed problems:
1. Since "Sensors MDPI" is not a seismological journal, so the abstract written for the general reader of "Sensors" requires 1-2 sentences introducing the specific seismological problem of "phase detection". For example, the abstract may start from: "When recording seismic ground motion in multiple sites using independent recording stations one needs to recognize the presence of the same parts of seismic waves arriving at these stations. This problem is known in seismology as seismic phase picking..." - or similar.  

2. reference no. 1 - correct volume or year

Author Response

Thank you for detailed comments and proposed editing! Please see the new revision.

Reviewer 2 Report

The manuscript "The Seismo-Performer: general and efficient machine learning approach for seismic phase recognition of local earthquakes in real time" presents an interesting machine learning application for seismic phase picking.

I have only some minor concerns:

1) Authors should  avoid abbreviations in the abstract.

2) They should write the complete meaning of the abbreviations the first time they appears in the text. In the manuscript, there are some abbreviations without any information about their meaning. 

Maybe, a summary table with all the abbreviations used in the manuscript will be helpful to the readers.

3) Page 3, line 117. I have a doubt. Is the dimension D used in equation (1) the projection dimension (see Table 1) or is it the dimension of the hidden space (see  page 4, line 142) ?. It should be explained the first time  the variable D is introduce in the text. 

4) Page 4, line 144. It seems that the number of hidden layers is 92 units, doesn't it? However, in Table 1, the parameter MLP hidden layer size indicates a value of 96. Please, clarify this issue.

5) Page 4, line 140 What is the Softmax activation function ? It is a function or your code, you should explain briefly what is the aim of this function.

6) Figure 3. I suppose that the values a and b are the Gütenberg-Richter parameters, aren't they?. They should be commented in the figure caption.

7) Page 7, line 232. You should explain what do you mean with the parameters precision / recall curves? How do you obtain the precision (and recall) of the proposed method? 

8) Table 3. Do these values represent the mean deviation (+- standard deviation) of the estimated P and S picking respecting to the manual picking? How do you calculate these accuracy values? What are the units of these values? miliseconds, seconds? 

9) Figure 4. What are the units of the axis variables?

10) Figure 4 and 5 captions. Pentagrams? Do you mean the red stars?

11) Figure 5. In the vertical axis of the model's prediction, what do the labels N, S and P mean?

12) Figure 5. With such a good SNR signal, even the simplest (and faster) methods will work fine. Maybe, the problem is when you have noisy signals from local low-magnitude events.

Author Response

First of all, thank you very much for carefully reading our text. Your review report helped us improve this article.

1) Authors should  avoid abbreviations in the abstract.

Fixed in new revision.

2) They should write the complete meaning of the abbreviations the first time they appears in the text. In the manuscript, there are some abbreviations without any information about their meaning. 

Fixed in new revision.

3) Page 3, line 117. I have a doubt. Is the dimension D used in equation (1) the projection dimension (see Table 1) or is it the dimension of the hidden space (see  page 4, line 142) ?. It should be explained the first time  the variable D is introduce in the text. 

Projection dimensions and hidden space are actually the same. Fixed in new revision.   4) Page 4, line 144. It seems that the number of hidden layers is 92 units, doesn't it? However, in Table 1, the parameter MLP hidden layer size indicates a value of 96. Please, clarify this issue.   The number of hidden layers is 96 units. Fixed in new revision. Thank you so much for that!   6) Figure 3. I suppose that the values a and b are the Gütenberg-Richter parameters, aren't they?. They should be commented in the figure caption.   Fixed in new revision. Thank you!   7) Page 7, line 232. You should explain what do you mean with the parameters precision / recall curves? How do you obtain the precision (and recall) of the proposed method?    Fixed in new revision.   8) Table 3. Do these values represent the mean deviation (+- standard deviation) of the estimated P and S picking respecting to the manual picking? How do you calculate these accuracy values? What are the units of these values? miliseconds, seconds?    Fixed in new revision.   9) Figure 4. What are the units of the axis variables?   It is dimensionless quantity from 0 to 1. It can be converted to percentages, but the current version of the plot is more common from our point of view. In addition, we have added Equations 6 and 7 with details of how these values were obtained.   10) Figure 4 and 5 captions. Pentagrams? Do you mean the red stars?   Fixed in new revision.   11) Figure 5. In the vertical axis of the model's prediction, what do the labels N, S and P mean?   12) Figure 5. With such a good SNR signal, even the simplest (and faster) methods will work fine. Maybe, the problem is when you have noisy signals from local low-magnitude events.   You are absolutely right. But for the demonstration, we have chosen a well-known earthquake for the graph. If we choose a weak earthquake, then for non-seismologist readers there will not be such an obvious distinct P / S phase.